# The association between *Dioscorea sansibarensis* and *Orrella dioscoreae* as a model for hereditary leaf symbiosis

Tessa Acar[1,2], Sandra Moreau[1], Marie-Françoise Jardinaud[1], Gabriella Houdinet[1], Felicia Maviane-Macia[1], Frédéric De Meyer[2], Bart Hoste[2], Olivier Leroux[3], Olivier Coen[1], Aurélie Le Ru[4], Nemo Peeters[1], Aurelien Carlier[1,2]*

1 LIPME, INRAE, CNRS, Université de Toulouse, Castanet-Tolosan, France, 2 Laboratory of Microbiology, Ghent University, Ghent, Belgium, 3 Department of Biology, Ghent University, Gent, Belgium, 4 Plateforme Imagerie TRI-FRAIB, CNRS, Université de Toulouse, Castanet-Tolosan, France

* aurelien.carlier@inrae.fr

**Data Availability Statement:** The datasets generated and/or analyzed during the current study are available in the recherche.data.gouv.fr public

## Abstract

Hereditary, or vertically-transmitted, symbioses affect a large number of animal species and some plants. The precise mechanisms underlying transmission of functions of these associations are often difficult to describe, due to the difficulty in separating the symbiotic partners. This is especially the case for plant-bacteria hereditary symbioses, which lack experimentally tractable model systems. Here, we demonstrate the potential of the leaf symbiosis between the wild yam *Dioscorea sansibarensis* and the bacterium *Orrella dioscoreae* (*O. dioscoreae*) as a model system for hereditary symbiosis. *O. dioscoreae* is easy to grow and genetically manipulate, which is unusual for hereditary symbionts. These properties allowed us to design an effective antimicrobial treatment to rid plants of bacteria and generate whole aposymbiotic plants, which can later be re-inoculated with bacterial cultures. Aposymbiotic plants did not differ morphologically from symbiotic plants and the leaf forerunner tip containing the symbiotic glands formed normally even in the absence of bacteria, but microscopic differences between symbiotic and aposymbiotic glands highlight the influence of bacteria on the development of trichomes and secretion of mucilage. This is to our knowledge the first leaf symbiosis where both host and symbiont can be grown separately and where the symbiont can be genetically altered and reintroduced to the host.

## Introduction

Heritable symbioses are permanent associations between two or more partners where at least one partner is directly (or vertically) transmitted to the next generation [1]. Often, species involved in heritable symbioses evolve a form of co-dependency, a phenomenon known as Muller's ratchet, that can result in hosts and symbionts becoming inseparable [1]. Heritable symbioses can be found throughout the tree of life, and are especially common in invertebrates [2–4]. Plants commonly engage in horizontally-transmitted symbioses, with established model systems such as the *Medicago- Sinorhizobium* symbiosis contributing to a better understanding

archive under https://doi.org/10.57745/R0VPGY (Phenotyping data of plants used to generate Fig 3, S2 Fig and S3 Fig).

**Funding:** This work was supported by the UGent Special Research Fund under grant BOFSTA2017002001 to AC. AC also acknowledges support from the French National Research Agency under grant agreement ANR-19-TERC-0004-01 and from the French Laboratory of Excellence project "TULIP" (ANR-10-LABX-41; ANR-11-IDEX-0002-02). The funders had no role in study design, data collection and analysis, decision to publish, or preparation of the manuscript.

**Competing interests:** The authors have declared that no competing interests exist.

of the mechanisms underlying nitrogen-fixing root nodule symbiosis [5]. However, there are few well-characterized hereditary associations between plants and bacteria, and the mechanisms enabling transmission and/or partner specificity are mostly unknown. In angiosperms, phyllosphere symbioses have been identified or suspected in the Rubiaceae, Primulaceae, Styracaceae and Dioscoreaceae families [6]. In particular, symbioses in *Ardisia* (Primulaceae), *Psychotria* (Rubiaceae) and *Pavetta* (Rubiaceae) have been relatively well-studied [7–10]. The function and transmission of leaf symbiosis are not well understood, but the shoot tip has long been suspected to be an important structure in leaf symbiosis. In leaf-nodulated Rubiaceae and Primulaceae species, a colony of obligate symbiotic bacteria residing near the apical meristem may serve as the source of infection for every new developing leaf and ovary, and thus the seeds [11–13]. Removal of bacterial symbionts from host plants in heritable leaf symbiosis has been studied extensively, and often leads to a stunted phenotype and death [14–16]. More recently, Sinnesael *et al.* showed that it was possible to grow the leaf-nodulated *Psychotria umbellata* without its *Candidatus* Caballeronia sp. symbiont *in vitro*, but aposymbiotic plants did not survive in soil [17]. Despite a sizeable body of work on leaf symbiosis in the Primulaceae and Rubiaceae families, plants are difficult to maintain due to long generation times, and bacterial symbionts are usually unculturable and genetically intractable [7–9, 17–22]. Because symbiotic bacteria of *Psychotria* and *Ardisia* cannot be cultured and host development is dependent on symbiotic status, many questions about transmission, function and the mechanisms underlying the specificity of leaf symbiosis remain unanswered. In contrast, *Orrella dioscoreae*, the bacterial symbiont of *Dioscorea sansibarensis*, has been isolated from leaves and is a notable exception [23, 24].

*D. sansibarensis* is the only monocotyledonous plant known so far to engage in leaf symbiosis, although related species may host similar epiphytes [25, 26]. The species likely originates from Madagascar and continental Africa and is invasive in parts of the US and South-East Asia [27]. In *D. sansibarensis*, the perennial vine thrives in hot and humid conditions and reproduces dominantly via bulbils (round, vegetative structures 2–3 cm in diameter) and tubers [28]. A single leaf gland forms at the acumen of the leaf and contains a dense mass of bacteria [29]. The *D. sansibarensis* leaf gland, also called forerunner tip, forms by folding of the lamina, resulting in hollow channels which subsequently fill with bacteria [30, 31]. Trichomes emerging from the epidermis protrude into the lumen of the glands and seem to be an important site for the symbiotic interaction. The function of the symbiosis remains unknown, although nitrogen fixation has been ruled out [30]. The bacterial symbiont was recently identified as *Orrella dioscoreae* (*O. dioscoreae*) and in contrast to most leaf symbionts, can be isolated and cultivated outside its host [23, 24]. Furthermore, the ease of culture, lack of resistance to antibiotics, and amenability to transformation by electroporation or conjugation make *O. dioscoreae* an attractive model system to understand the functions required for the endophytic lifestyle of leaf symbiotic bacteria [23, 24].

Establishing the *D. sansibarensis*/*O. dioscoreae* as an experimental model requires manipulating the symbiotic status of the plant. Because pathogen-free plants are of high interest for the horticulture industry, several methods have been developed to control fungal and bacterial contaminants in plants or tissue culture [32]. Seed surface sterilization is a popular technique used in crops and *Arabidopsis thaliana* to remove pathogens from seeds [33–36]. This is done by treating seeds with solutions of sodium hypochlorite and/or ethanol, but surface treatment is often insufficient to rid the seeds of endophytic microorganisms, which are presumably embedded in plant tissue out of reach of disinfectants [37–40]. To remove recalcitrant contaminants, more effective methods make use of tissue culture followed by regeneration of whole plants. For example, plant structures containing meristematic cells (e.g. buds or embryos) may be isolated and grown under sterile conditions with auxins and/or cytokinins to promote

cellular growth and differentiation [41–44]. This type of vegetative propagation combined with heat treatment is effective for clearing some viruses from germplasms [45–47], but may lack efficacy against fungal or bacterial endophytes. Antibiotics are an effective mean of clearing bacteria and fungi, but plant tissue cultures are often susceptible to damage from some commonly used antibiotics [48]. However, β-lactam antibiotics such as cefotaxime or carbenicillin are well tolerated by wheat tissue culture [49] and fungal contamination may be controlled using carbendazim, fenbendazole and imazalil [50]. In this study, we tested and developed an effective series of protocols to obtain aposymbiotic *D. sansibarensis.* Aposymbiotic plants developed normally under controlled conditions, and could be inoculated by exogenous *O. dioscoreae* strains using simple methods. Altogether, these properties make the *Dioscorea-Orrella* symbiosis an appealing candidate for a heritable leaf symbiosis model system.

## Material and methods

### Plant culture and propagation

*Dioscorea sansibarensis* Pax plants were obtained from the greenhouse of the Botanical Garden at the University of Ghent (LM-UGent) in Ghent, Belgium. Chemicals and reagents were purchased from Merck unless otherwise indicated. Plants used throughout in experiments were maintained in the greenhouse of the Laboratory of Plants Microbes and Environment Interactions (LIPME) in Castanet-Tolosan, France. Unless otherwise indicated, plants were grown in climate chambers at 28°C, 70% humidity and a light cycle of 16h light (210 μmol/m$^2$/s), 8h dark.

### Bacterial strains and culture conditions

*O. dioscoreae* strains were grown in tryptic soy agar (TSA) or broth (TSB) aerobically at 28°C unless specified otherwise. Media were supplemented with gentamicin (20 μg/mL) and/or nalidixic acid (30 μg/mL) as appropriate. *O. dioscoreae* strain R-71412 is a spontaneous nalidixic acid-resistant strain derived from *O. dioscoreae* LMG 29303[T] [24]. *O. dioscoreae* strains R-71416 and R-71417 are gentamycine-resistant derivatives of strain R-71412 with a chromosomally-encoded *gfp* or *mCherry* reporter genes, respectively [31]. *O. dioscoreae* strains R-67173, R-67584, R-67088 and R-67090 are natural isolates described in a previous publication [23]. Details are available in S1 Table.

### Surface sterilization and inoculation of bulbils

Inoculation of bulbils by bacterial submersion was done as follows: bulbils were peeled and sterilized in 0.15% carbendazim for 2 hours, washed 3 times with sterile water, submerged in ethanol (70% v/v) for 5 minutes, transferred to sodium hypochlorite (1.4% v/v) + 0.4% v/v Tween 20 for 15 minutes and washed 3 times with sterile distilled water. Bulbils were incubated in half-strength Murashige and Skoog basal medium (MS) supplemented with gelzan (4 g/L) at 28°C in sterile Microbox containers (Sac O$_2$, Belgium) with a 16h/8h photoperiod. Bulbils were inoculated with *O. dioscoreae* R-71416 (S1 Table) as follows: bacterial cultures were grown in Tryptic Soy Broth (TSB) to exponential phase, centrifuged (7500 rpm, 10 min) and washed twice with sterile 0.5x Phosphate buffered saline (0.5x PBS: 4 g/L NaCl, 0.1 g/L KCl, 0.72 g/L Na$_2$HPO$_4$, 0.12 g/L KH$_2$PO$_4$, pH 7.4). Cell suspensions were normalized to OD$_{600nm}$ = 0.2 and bulbils were submerged in 50 ml bacterial suspension for 3 hours while shaking (100 rpm) at room temperature. Bulbils were then placed in sterile Microbox containers (Sac O$_2$, Belgium) with 50 ml of half-strength MS medium supplemented with Gelzan 4g/L and

incubated at 28˚C, and a 16h/8h photoperiod. Alternatively, *O. dioscoreae* cell suspensions prepared as above were injected directly into surface-sterilized bulbils with a 26G needle at a randomly chosen site. Bulbils were incubated in Microboxes as stated above.

## Detection and identification of bacteria

The tip of the leaf was dissected with sterile tweezers and a scalpel, and the tissue was homogenized using 100 μl 0.4% w/v NaCl and 3 sterile glass beads for 1 minute at 30 Hz in a ball mill (Retsch MM 400). The homogenized suspension was centrifuged briefly to pellet debris. One hundred μL of supernatant was directly plated out on Tryptic Soy Agar (TSA) plates and incubated for 2 days at 28˚C. If the plate showed growth, one isolate per colony type was picked and identified using colony PCR with primers specific for O. *dioscoreae* (nrdA-01-F: `GAACTG GATTCCCGACCTGTTC`, nrdA-02-R: `TTCGATTTGACGTACAAGTTCTGG`), or with universal 16S rRNA primers (pA: `AGAGTTTGATCCTGGCTCAG` and pH: `AAGGAGGTGATCCAGCCGCA`) followed by Sanger sequencing.

## Direct inoculation of shoot tips

Plants were grown from sterilized bulbils in sterile conditions until emergence of the shoot. The shoot tip was sprayed with gentamycin dissolved in water (20 mg/ml, Méridis France). Plants were inoculated with bacteria as follows: bacterial cultures grown in TSB to about $OD_{600nm} = 0.5$ were centrifuged (7500 rpm, 10 min) and washed twice with sterile 0.5x PBS. Cell suspensions were normalized to $OD_{600nm} = 0.2$. Different methods were used to inoculate the shoot tip. A- The biggest leaf at the apical bud was gently pushed aside and a small scratch was made on the apical bud with a 27 G needle. The apical shoot tip was dipped in the bacterial suspension ($OD_{600nm} = 0.2$) for 15 seconds (dippling method). B- The apical bud was stabbed with a tuberculin needle dipped in the bacterial suspension. Sonicating: this methods is adapted from a protocol designed for agroinfiltration using sonication [51]. Briefly, dissected apical buds were submerged in a bacterial suspension in an 2 mL microfuge tube and placed in a Branson Ultrasonic 2800 sonication bath using a floating device for 2 minutes (Stabbing method). C- Dipping the apical bud in a liquid bacterial suspension of strain R-71416 followed by vacuum infiltration [52] in a dessicator maintained at 0.53 bar for 2 minutes (Vacuum infiltration method). All plants were put in sterile microboxes in a 1:1 (v/v) pumice/perlite mixture at 28˚C, 16h/8h photoperiod.

## Minimal inhibitory concentrations assay on *O. dioscoreae*

Liquid cultures grown in TSB (R-67173, R-67584, R-67088, R-67090 and LMG 29303$^T$) in exponential phase were diluted to $OD_{600nm} = 0.001$ (~$10^6$ CFU/ml). Serial dilutions of antibiotics were prepared in sterile water (1024-512-256-128-64-32-16-8 μg/ml) and liquid cultures were added in a 1:1 ratio to the antibiotic solution. Samples were well mixed and incubated at 28˚C for 48 hours.

## Propagation through node cuttings and antibiotic treatment

Micropropagation of *D. sansibarensis* was done using a protocol adapted from [53]. Node cuttings were collected from greenhouse-grown plants 2–4 months after emergence. Two distinct protocols were used for surface sterilization. In the 'bleach + ethanol' protocol, explants were first washed with tap water, surface sterilized by submerging for 2 hours in a sterile solution of 0.15% w/v carbendazim + 0.4% v/v Tween 20, washed 3 times with sterile distilled water, then soaked in 70% v/v ethanol for 5 minutes, and finally 1.4% w/v sodium hypochlorite + 0.4% v/v

Tween 20 for 15 minutes. Explants were then washed 3 times in sterile distilled water). Alternatively, fresh explants were soaked in 3 x concentrated MS medium supplemented with 5% (v/v) solution of Plant Preservative Mixture (PPM, Plant Cell Technology, USA) with shaking at 100 rpm for 8 hours at 28˚C ('PPM protocol'). PPM is a commercial biocide containing the active ingredients 5-chloro-2-methyl-3(2H)-isothiazolone and 2-methyl-3(2H)-isothiazolone, which can be supplemented directly to the culture medium [54]. After 8 hours, the bleached extremities of the explants were cut off with a sterile scalpel. Explants were placed in sterilized growth medium (MS basal medium 4.4 g/L, 2% w/v sucrose, vitamins: glycine (2 mg/L), myo-inositol (100 mg/L), nicotinic acid (0.5 mg/L), pyridoxine-HCl (0.5 mg/L), thiamine-HCl (0.1 mg/L) and L-cystein (20 mg/L), pH = 5.7), supplemented with the antibiotics carbenicillin (200 µg/ml), cefotaxime (200 µg/ml) and PPM (0.2% v/v) and incubated at 28˚C, 16h/8h photoperiod. Medium was refreshed after 10 days, including supplements and antibiotics. After 21 days of incubation, the medium was replaced with growth medium containing MS, sucrose, PPM and vitamins as described above but without the antibiotics. Cuttings were transferred in sterile Magenta boxes (model GA7, Merck) incubated at 28˚C, 16h/8h photoperiod.

## Transmission electron microscopy (TEM)

Samples were fixed in 2% w/v glutaraldehyde + 0.5% w/v paraformaldehyde (v/v) in a 50 mM sodium cacodylate buffer, pH 7.2 at room temperature and under vacuum. After 4 hours, the fixative solution was refreshed and samples were kept at 4˚C for 26 days. Samples were rinsed twice in 50 mM sodium cacodylate buffer (pH 7.2) and postfixed in 2% w/v osmium tetroxide in water for 1.5 hours at room temperature. Samples were rinsed 3 times in demineralized water and dehydrated using a graded water/ethanol series (10, 20, 30, 40, 50, 60, 70, 80, 90, 96% (v/v)). Samples were first incubated in propylene oxide (Electron Microscopy Sciences, Hatfield PA, USA) twice for 1 hour, then in a propylene oxide /Epon series over several days at 4˚C, positioned in their silicone embedding molds and polymerized for 48 hours at 60˚C. Thin sections were cut using a Reichert Ultracut E (Leica Microsystems) and contrasted using Uranyless and lead citrate (Delta Microscopies, France). Samples were observed using a Hitachi HT7700 instrument.

## Scanning electron microscopy (SEM)

Samples were fixed in 2.5% v/v glutaraldehyde in 50 mM cacodylate sodium buffer (pH 7.2) for 3 hours at room temperature and transferred to 4˚C for 2 days. They were dehydrated using a graded water/ethanol series (10, 20, 30, 40, 50, 60, 70, 80% (v/v)). The samples were completely dehydrated using a critical point drying apparatus (Leica EM CPD 300) using $CO_2$ as transitional medium, and a platinum coating was applied. Samples were examined using a FEG FEI Quanta 250 instrument.

## Light microscopy

Samples were fixed in 4% v/v formaldehyde in PEM buffer (100 mM 1,4-piperazinediethane-sulfonic acid, 10 mM $MgSO_4$, and 10 mM ethylene glycol tetra-acetic acid, pH 6.9) for 4h, thoroughly washed in PBS and dehydrated using a graded ethanol series (30, 50, 70, 85, 100% v/v). After gradual infiltration with LR White acrylic resin (medium grade, London Resin Company, UK), samples were embedded in polypropylene flat bottom molds at 37˚C for 3 days. Semi-thin sections of 300 nm, cut using a Leica UC6 ultramicrotome equipped with a diamond knife, were dried onto polysine-coated slides, stained with 1% w/v toluidine blue in 0.5% w/v sodium tetraborate for 5 seconds and mounted in DePeX (VWR, Belgium). For vibratome sectioning, samples were embedded in 8% w/v agarose, glued upon the specimen

stage using Roti coll 1 glue (Carl Roth, Karlsruhe, Germany) and cut into 30 μm thick sections with a vibrating microtome (HM650V, Thermo Fisher Scientific, Waltham, MA, USA). Sections were stained in 0.5% w/v astra blue, 0.5% w/v chrysoidine and 0.5% w/v acridine red for 3 min, rinsed with demineralized water, dehydrated with isopropyl alcohol and mounted in Euparal (Carl Roth, Karlsruhe, Germany). Vibratome and LR White sections were observed using a Nikon Eclipse Ni-U bright field microscope equipped with a Nikon DS-Fi1c camera. To visualize mCherry tagged *O. dioscoreae* (R71417) in the shoot tips, fresh plant samples were hand cut and directly observed by confocal microscopy (Leica TCS SP2) using excitation wavelength of 552 nm and emission collection between 584–651 nm. GFP-tagged bacteria were visualized using excitation at 488 nm and emitted light from 500 to 550 nm. Leica LAS X software was used to process the images.

## Inoculation of aposymbiotic *D. sansibarensis* with bacteria

Node cuttings were grown in axenic conditions (25ml MS + 2% w/v sucrose + 0.2% v/v PPM in Magenta vessel, 28˚C, 16h/8h photoperiod) until a new shoot appeared (after 6 weeks approximately). Verified aposymbiotic plants (tested as above) were inoculated with a strain of interest as follows: bacterial cultures in the exponential phase of growth were centrifuged (5000 rpm, 10 min) and washed twice with sterile 0.4% w/v NaCl. Cell suspensions were normalized to $OD_{600nm}$ = 0.2. The biggest leaf at the apical bud was gently pushed aside and 2 μl of a bacterial suspension (corresponding to approximately 5 x $10^6$ CFU) was deposited onto the apical bud (S1 Fig). Plants were transferred to sterile Microbox containers (50 ml MS + 2% w/v sucrose + 0.2% v/v PPM) at 28˚C, 16h/8h photoperiod cycles until new leaves emerged. Colonization was evaluated by dissecting a leaf tip and spreading the contents on suitable microbiological medium as described above (Detection and identification of bacteria).

## Plant phenotyping

Plants were grown from node cuttings in axenic conditions in Magenta boxes containing and inoculated with *O. dioscoreae* strain R-71412 or a sterile solution of 0.4% w/v NaCl as described above. Plants were kept in gnotobiotic conditions in Microbox containers containing (50 ml MS + 2% w/v sucrose + vitamins + 0.2% w/v PPM) at 25˚C, with a 16h/8h photoperiod. Pots were randomly distributed and shuffled once a week during the experiment. Plants were collected 4 weeks post-inoculation. Leaves were separated from the stem by cutting the petioles with a scalpel, and photographed using a ruler for scale. Chlorophyll content, nitrogen balance index, anthocyanins index and epidermal flavonols were measured on the leaf lamina at 2 different spots immediately after detaching, using a Dualex optical leafclip meter (Force-A, Orsay, France). Stem length was measured with a ruler from crown to tip. Leaf length, width, area and acumen length were determined from photographs using the Fiji software [55]. To control for developmental stage, the position of each leaf relative to the shoot tip was recorded for each plant, with leaf n˚1 being the closest from the shoot tip, excluding currently emerging leaves. The experiment was repeated twice independently in the same growth chamber. All statistical analyses were done in R using the standard 2-sided Wilcoxon rank sum test (wilcox.test) function [56].

## Automated plant phenotyping in greenhouse conditions

Twenty-five plants obtained from node cuttings and grown for 6 weeks in gnotobiotic conditions were transferred to soil in 3L pots in a climate-controlled greenhouse at 25˚C, 60% humidity and a light cycle of 16h light (179 μmol/m$^2$/s), 8h dark. A blue foam disc was placed on top of the pot to increase contrast for image segmentation, and a blue-colored plastic cage

was placed in the pots to guide plant development. The symbiotic status of the plants was checked as described above and aposymbiotic plants were inoculated with a Mock solution (0.4% (m/v) NaCl) or a liquid culture (LMG 29303$^T$) as described above, after 2 leaves had emerged. As plants grew at different paces, plants were inoculated on different dates at the 2 leaf stage. To monitor the symbiotic status of the plants, samples from leaf glands were taken at 3 different timepoints during the experiment. Plants grown from node cuttings were tested for the presence of *O. dioscoreae* in mature leaf glands, with 7 out of 25 plants still harboring *O. dioscoreae* (S3 Table). One aposymbiotic plant died at the beginning of the experiment, and 3 others went into growth arrest in the following days, possibly due to stress from the transfer from sterile containers to open pots. Of the 14 aposymbiotic plants remaining, half were inoculated with strain LMG 29303$^T$ and half with a mock solution. After 30 days, the height of the stem and the number of leaves were measured and counted. Plant development was monitored automatically for at least 30 days after inoculation in the Phenoserre facility of the Toulouse Plant-Microbe Phenotyping platform (TPMP) and their symbiotic status was checked 3 times by isolation of bacteria from leaf glands and PCR as described above. Each plant was imaged once a day using and RGB camera and a blue background, rotating the plant at 6 angles (0˚ to 300˚ in 60˚ increments). Image analysis was done using the IPSO Phen software v1.20.3.17 (https://github.com/tpmp-inra/ipso_phen) [57], resulting in a total of 56 parameters measured, including 37 measures of morphology, e.g. total area, hull, width, height. Additional parameters linked to colorimetry, including mean and standard deviation for all channels in various color spaces (RGB, LAB and HSV) were also recorded. As *D. sansibarensis* vines tended to grow in irregular patterns, no morphological parameters could be reliably analyzed except for total leaf area, which was calculated as the median of leaf area extracted of images from all 6 angles. Chlorophyll content was estimated through RGB values of plant images as described by Liang and colleagues [58]. Plants were automatically watered daily and fertilized at the beginning and once mid-experiment. 30 days after of the last inoculation, the length of the stem and the number of leaves were measured by hand. All statistical analyses were done in R with the wilcox.test or kruskal.test functions [56].

## Results

### Symbiotic *D. sansibarensis* are recalcitrant to inoculation with exogenously applied *O. dioscoreae*

To investigate if symbiotic structures remained open to colonization, we first attempted to introduce fluorescent-tagged *O. dioscoreae* in wild-type symbiotic *D. sansibarensis*. Because *D. sansibarensis* rarely flowers in cultivation, we attempted to inoculate aerial bulbils with suspensions of *Orrella dioscoreae*. Submerging whole bulbils in a suspension of *O. dioscoreae* R-71416 did not result in colonization of germinated seedlings by GFP-tagged bacteria, as evidenced by a lack of fluorescent colonies when macerates were plated on selective media. Bulbils have a suberized outer tissue layer, which might prevent exogenous bacteria from reaching the vegetative growth center. To test this, we peeled and surface-sterilized 6 bulbils, which we submerged in a suspension of GFP-tagged *O. dioscoreae* R-71416. As control, 3 bulbils were submerged in sterile saline solution and left to germinate. Every bulbil deteriorated and failed to yield new plants. We also attempted to deliver a bacterial inoculum in 5 surface-sterilized, unpeeled bulbils by injection with a needle. The bulbils germinated, but only non-fluorescent colonies could be isolated from macerates, indicating that only wild-type *O. dioscoreae* could be recovered from the leaf glands of the plantlets.

We hypothesized that inoculating the shoot tip with bacteria would result in colonization of all shoot tissue growing from the apical meristem. We dipped shoot tips in a suspension of *O.*

*dioscoreae* R-71416 and macerated the newly emerged leaves. Leaf glands always contained only wild-type non-fluorescent *O. dioscoreae*. Stabbing the apical shoot tip with a needle dipped in a bacterial suspension resulted in 4 out of 4 shoot tips turning necrotic within days. Vacuum infiltration of a liquid bacterial suspension of strain R-71416 into shoot tips resulted in growth arrest of the 4 plants tested. Two plants formed bulbils, but we could not detect growth of *O. dioscoreae* R-71416 in macerates. Finally, we attempted to inoculate the plants by adapting a protocol designed for agroinfiltration using sonication. Of the 3 plants tested, one plant went into growth arrest, but *O. dioscoreae* R-71416 could not be detected in leaves of any of the remaining plants.

## Treatment of node cuttings with an antibiotic cocktail results in aposymbiotic plants

We reasoned that processes such as competition and niche exclusion might contribute to preventing exogenous GFP-tagged *O. dioscoreae* from infecting already symbiotic plants. Miller and Reporter previously described the generation of aposymbiotic plants from surface-sterilized bulbils of *D. sansibarensis* [30]. We attempted to reproduce these results by surface-sterilizing bulbils and incubating in sterile Microbox containers containing sterile medium. All bulbils germinated, but leaf glands of 18/18 plants contained *O. dioscoreae*, showing that surface sterilization alone was not sufficient to create aposymbiotic plants. Next, we adapted a protocol used to micropropagate the yam species *Dioscorea composita*, to which we added an antibiotic treatment. We first tested the susceptibility of *O. dioscoreae* strains to antibiotics commonly used in plant tissue culture. All *O. dioscoreae* strains were sensitive to tetracyclin and rifampicin (minimum inhibitory concentrations < 16 μg/ml); and moderately resistant to the β-lactam antibiotics carbenicillin and cefotaxime (S2 Table). All strains were also sensitive to the commercial broad-range biocide Plant Preservation Mixture (PPM). All antibiotics tested inhibited growth of the *O. dioscoreae in planta*, but tetracycline and rifampicin also impaired plant growth at the concentrations tested (Table 1). Only carbenicillin and cefotaxime at concentrations of up to 200 μg/mL were effective against *O. dioscoreae* and were well tolerated by plant tissue (Table 1). Although incubation with antibiotics was effective to remove *O. dioscoreae* from node cuttings, approximately half of our *in vitro* cultures were lost to contamination of the tissue and media with fungi and bacteria. We reasoned that incomplete surface-sterilization of bulbils may be a source of contaminants and we tested treatment with the commercial biocide PPM to control microbial contamination in *in vitro* cultures of *D. sansibarensis*. According to the manufacturer, PPM may also be used as a mild antiseptic for surface sterilization of plant tissue. Surface sterilization with a solution of 5% v/v PPM in 3x MS medium for 8h at 28°C in darkness with shaking was sufficient to prevent contamination

**Table 1. Effect of different antibiotics on the *in vitro* growth of *D. sansibarensis* and removal of its bacterial symbiont *O. dioscoreae*.**

| Antibiotic | Concentration | Contact Time | Effect on plant growth | *O. dioscoreae* cfu/explant |
|---|---|---|---|---|
| Carbenicillin + cefotaxime | 100 μg/ml | 1 week | No effect | $< 10^2$ |
| | | 3 weeks | No effect | 0 |
| | 200 μg/ml | 1 week | No effect | $<10^3$ |
| | | 3 weeks | No effect | 0 |
| Tetracycline | 50 μg/ml | 1 week | Explant ends turn brown. No growth. | $<10^3$ |
| | | 3 weeks | Explant ends turn brown. No growth. | 0 |
| Rifampicin | 200 μg/ml | 1 week | Explant ends turn brown. No growth. | $<10^2$ |
| | | 3 weeks | Explant ends turn black. Few emerging leaves are chlorotic. | 0 |

**Table 2. Efficiency comparison between node cutting sterilization protocols.**

| | BLEACH + ETHANOL PROTOCOL | PPM protocol |
|---|---|---|
| **Number of plants treated** | 89 | 105 |
| **Number of visibly contaminated cultures** | 34 (38.29%) | 12 (10.17%) |
| **Number of dead explants** | 5 (5.62%) | 0 (0%) |
| **Aposymbiotic plants** | 18 (31.12%) | 47 (49.33%) |

while preserving tissue viability (n = 18). Using the PPM protocol, 0/105 cuttings were lost to death of the explant, while 5/89 cuttings were lost using the bleach + ethanol protocol (Table 2). In the first 3 weeks of incubation with antibiotics, 34 plants were lost due to contamination with the bleach + ethanol protocol (38.3%), while only 12 plants (10%), were lost using the PPM protocol. After 3 weeks, only 31% of resulting plantlets were aposymbiotic using the bleach + ethanol protocol, while 49.3% of node cuttings were aposymbiotic when sterilized with the PPM protocol.

## Microscopic differences between aposymbiotic and symbiotic *D. sansibarensis*

To investigate whether the loss of the symbiotic bacteria induces phenotypic or developmental changes, we generated plants through node cuttings using the "PPM" protocol as described above. Leaves of plants were tested after 6 weeks for the presence of *O. dioscoreae* in leaf glands. Aposymbiotic, as well as plants which remained symbiotic despite antibiotic treatment, were transferred to sterile containers and kept in sterile conditions without antibiotics. Leaves of aposymbiotic plants displayed fully-formed leaf glands, visually indistinguishable from those of the symbiotic plants (Fig 1A and 1B). Neither symbiotic nor aposymbiotic plantlets showed chlorosis or developmental abnormalities (Fig 1E and 1F). Microscopically, leaf glands of symbiotic plants were filled with bacteria embedded in extracellular matrix or mucus, with numerous trichomes projecting from the epithelium to the inside of the gland (Fig 1C). In contrast, aposymbiotic glands appeared somewhat flat, with no visible bacteria and fewer trichomes (Fig 1D). Cross-sections of leaf acumens imaged by scanning electron microscopy looked undistinguishable at low magnification (Fig 1G and 1H), but the lack of bacteria and mucus in aposymbiotic leaf glands became clear at higher magnification (Fig 1I and 1J). Trichomes were visible in both sample types, but only symbiotic samples contained bacteria (Fig 1K and 1L). Trichomes in aposymbiotic acumens appeared less electron-dense under the transmission electron microscope, with large vacuoles and sometimes visible loss of membrane integrity (Fig 1K and 1L). Golgi, vesicles and endoplasmic reticula (ER), components that suggest interaction between the host and the symbiont, were less abundant in aposymbiotic glands (Fig 1M and 1N).

## Symbiont replacement by drop-infection on aposymbiotic plants

We reasoned that aposymbiotic plants may be more amenable to colonization with exogenously applied bacteria. To test this, we inoculated 10 aposymbiotic *D. sansibarensis* kept in sterile containers with a 2 μL drop of a cell suspension of *O. dioscoreae* strain R-71417, which was deposited directly on the shoot apical bud (S1 Fig). All plants were successfully colonized, and 9 out of 10 plants had grown new leaves 3 weeks after inoculation (Fig 2). Up to 95% of our plants were successfully inoculated and the method rarely induced growth arrest in subsequent experiments. No bacteria could be found in leaf glands under the point of inoculation.

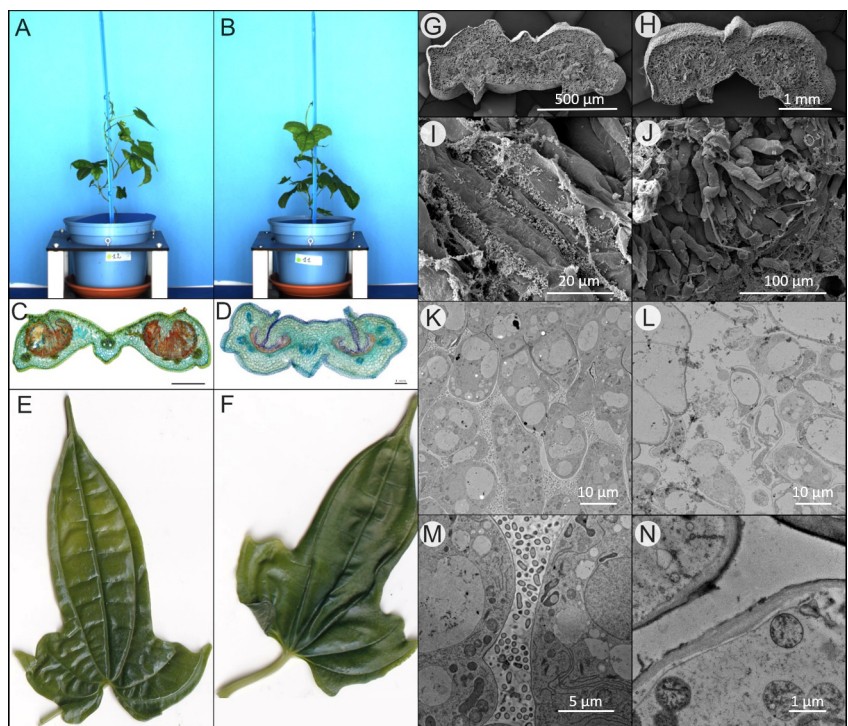

**Fig 1. Phenotypic differences between symbiotic (left) and aposymbiotic (right) *D. sansibarensis*. A.** Plants inoculated with *O. dioscoreae* or **B.** with a mock solution. **C.** Cross-section of *D. sansibarensis* gland stained with acridine red, chrysoidine and astra blue showing a dense, orange-colored mixture of mucus and bacteria filling the lumen of symbiotic glands, and **D.** glands of aposymbiotic plants; **E.** Adaxial side of leaves of symbiotic; **F.** aposymbiotic plants kept in gnotobiotic conditions. **G.** SEM cross-section of symbiotic and **H.** of aposymbiotic acumen. **I.** SEM detail picture of trichome cells in the acumen being colonized by bacteria or **J.** aposymbiotic. **K.** TEM of trichomes in the acumen, surrounded by bacteria in symbiotic glands or **L.** deteriorating in aposymbiotic glands. **M.** Close-ups TEM showing the endoplasmic reticulum, Golgi, and plastids in the trichomes; and **N.** being mostly empty and containing plastids.

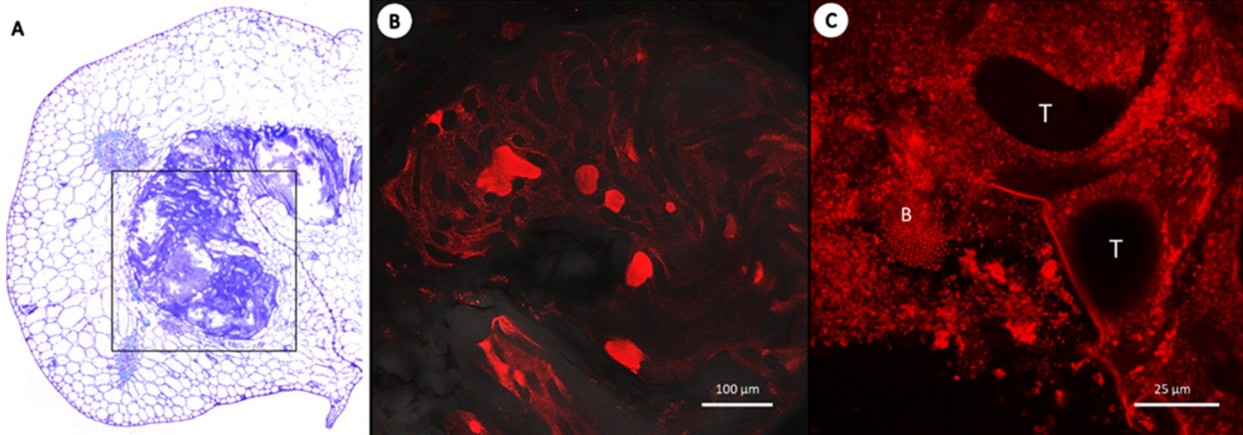

**Fig 2.** Fluorescence microscopy of the symbiotic gland at the acumen **A.** Overview of a TBO–stained transverse section viewed under brightfield, showing one gland in the leaf drip-tip. **B.** Close-up of a *D. sansibarensis* leaf gland colonized by mCherry-tagged bacteria (R-71417). **C.** Close up showing bacteria (B) surrounding the trichomes (T).

## Aposymbiotic *D. sansibarensis* develop normally under gnotobiotic conditions

To determine if the loss of symbiotic bacteria affected seedling growth and development, we inoculated aposymbiotic plants with cell suspensions of *O. dioscoreae* R-71412 or a sterile mock solution. After 4 weeks of growth in gnotobiotic conditions, we did not detect significant differences between aposymbiotic and re-inoculated plants for any of the morphological and physiological parameters we measured, including leaf area, length of the forerunner tip, stem length (S2 Fig) as well as chlorophyll, anthocyanins, flavonoids content and nitrogen nutritional status (S3 Fig).

## No phenotypic difference between aposymbiotic and symbiotic *D. sansibarensis* in the greenhouse

To follow development of aposymbiotic and symbiotic plants further in semi-natural conditions, we planted 24 PPM-treated plantlets into open pots filled with soil. These plants were tested after a short period of recovery, and 14 plants were certified aposymbiotic, while 10 still tested positive for bacteria in the leaf glands. Aposymbiotic plants were inoculated in the greenhouse in non-sterile conditions by shoot tip inoculation of a saline solution or a bacterial suspension as described above and continuously monitored for 52 days in a high-throughput plant phenotyping facility. Inoculation by dripping suspensions of *O. dioscoreae* on aposymbiotic shoot tips in the greenhouse was inefficient, with only 4/7 plants successfully inoculated (S3 Table). Unexpectedly, 4 plants which started out as aposymbiotic tested positive to *O. dioscoreae* and/or other bacteria in later stages of the experiment. In addition, 5 plants that tested positive for *O. dioscoreae* at the beginning of the experiment also produced bacteria-free leaf glands. Because of their uncertain status, these plants were labeled as "unknown status" in our analyses and treated as a third category. Although highly variable between individuals, we did not detect differences between symbiotic or aposymbiotic plants with regards to chlorophyll fluorescence (Fig 3A). Similarly, the number of leaves and length of the stems did not differ significantly between aposymbiotic, symbiotic and "unknown status" plants (Fig 3B and 3C).

## Discussion

We explore in this work the experimental tractability of the *D. sansibarensis*/*O. dioscoreae* association to answer fundamental questions about heritable symbiosis in plants. The ability to culture both partners separately and to manipulate infections is essential for the association to serve as an experimental model system for leaf symbiosis. Our initial attempts to introduce exogenous *O. dioscoreae* into symbiotic *D. sansibarensis* shoot apical buds or bulbils without first clearing the native symbionts were unsuccessful, and harsh inoculation techniques such as submerging, stabbing, or vacuum infiltration resulted in death or growth arrest of the plant. This indicates that exogenously-applied bacteria may be unable to reach the inside of the shoot tip, either due to host-derived barriers or spatial exclusion by resident *O. dioscoreae*. We developed an effective and reliable method to remove endophytic bacteria from shoot explants, specifically by treating node cuttings with an antibiotic cocktail, followed by *in vitro* regeneration of whole plants. Aposymbiotic plants obtained from explants treated with an antimicrobial cocktail and kept in sterile conditions were amenable to inoculation with exogenous *O. dioscoreae*, with high infection rates (>95%) from simply applying a bacterial suspension on shoot tips. However, aposymbiotic plants kept in sterile conditions are easily amenable to inoculation with exogenous bacteria, while plants in open pots in the greenhouse are more refractory. Together, these results suggest that prior infection with *O. dioscoreae* precludes other bacteria

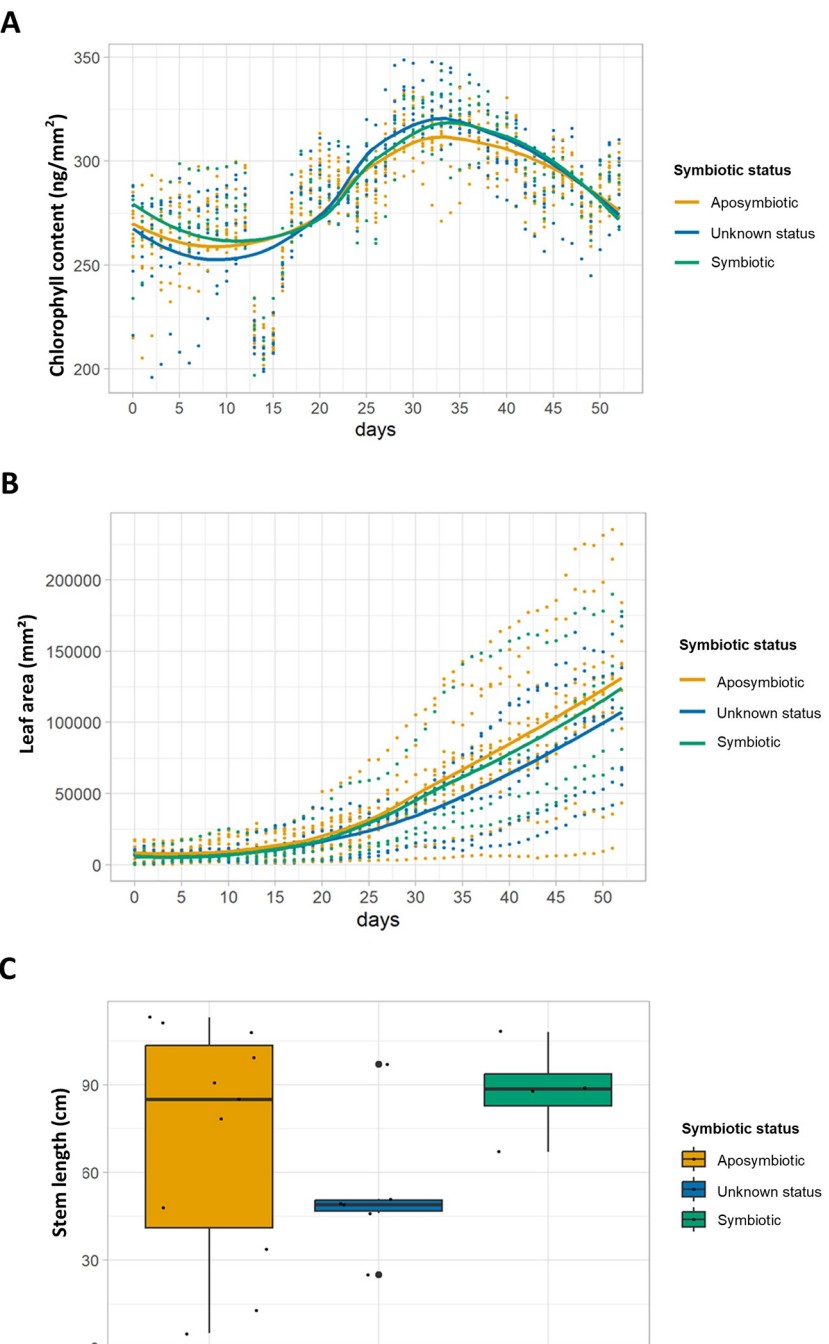

**Fig 3. Macroscopic phenotypes of aposymbiotic and symbiotic *D. sansibarensis*. A**. Daily mean chlorophyll content of individual plants tracked over a period of 30 days post inoculation estimated through RGB values of plant images. Trajectories of aposymbiotic plants are shown in yellow, symbiotic plants in green and plants with unknown status (see text for details) in blue. Solid lines depict the moving average per condition. Statistical testing was done with non-parametric Kruskal-Wallis tests, grouping samples by date (p-value for each test > 0.05). **B**. Total leaf area of individual plants tracked over a period of 30 days post inoculation. Color scheme is the same as above. Statistical testing was done as above (Kruskal-Wallis *p* > 0.05) **C**. Stem length (in cm) of plants measured at the end of the experiment. Data from aposymbiotic plants are shown in yellow, symbiotic plants in green and plants of unknown status in blue (see main text for details). Black dots are values randomly scattered around a central axis to improve readability. Large black dots in the "Unknown status" category represent distribution outliers. The distributions of values between the 3 categories of plants are identical for each of the 3 parameters (Kruskal-Wallis test p > 0.05).

from colonizing leaf glands. Whether this is due to bacteria-bacteria competition, antagonistic interactions, or a host response remains to be elucidated.

Aposymbiotic plants were also macroscopically indistinguishable from symbiotic plants. Both types of plants seemed healthy with no signs of chlorosis, with normal growth and development (Figs 1–3 and S2 Fig). Leaf glands that host bacteria in symbiotic plants were fully formed in aposymbiotic plants although they appeared somewhat thinner and less turgid than symbiotic glands (Fig 1). This is in contrast to leaf nodule symbiosis in the *Psychotria* genus, where aposymbiotic plants present few, abnormal or no leaf nodules [17]. Microscopically, glands of aposymbiotic leaves did not contain visible bacteria or copious amounts of mucus as with symbiotic plants. Whether this mucus is plant-produced, bacteria-produced or both is not known. Leaf glands differed in appearance from symbiotic ones. Overall, they showed fewer Golgi, ER and vesicles (Fig 1K–1N). Some aposymbiotic trichomes seemed atrophied, a phenotype also described in earlier work [30]. The fact that bacteria-free leaf glands formed normally in aposymbiotic *D. sansibarensis* offers attractive opportunities to investigate the host response to a symbiotic partner in this specialized organ.

Interestingly, symbiotic and aposymbiotic plants were phenotypically indistinguishable. We did not detect significant defects in plant development or photosynthetic functions between plants harboring *O. dioscoreae* or aposymbiotic controls. This is in stark contrast to leaf nodule symbiosis in *Ardisia crenata*, *Psychotria kirkii* (syn. *P. punctata*) and *Psychotria umbellata*, where loss of symbiotic bacteria is invariably linked to severe developmental defects and eventually death [13, 17, 20, 59]. This is also contrary to previous observations on the *Dioscorea* leaf symbiosis by Miller and Reporter [30]. These authors reported that the association between the plant and the (then unidentified) leaf gland bacteria was facultative, but bacteria-free plants were small and appeared chlorotic. This difference with our observations may be explained by the fact that Miller and Reporter grew plants from sterilized bulbils in sterile glass jars with seals that may affect gas exchange. These same authors also claim to have obtained bacteria-free plants by surface sterilization of bulbils with bleach and ethanol. Despite our best attempts to replicate their protocols, surface sterilization of bulbils never resulted in aposymbiotic plants in our hands. Our results suggest instead that *O. dioscoreae* does not play a major role in plant development. Previous analysis of the *O. dioscoreae* genome also ruled out a role in mineral nutrition, such as nitrogen fixation [23]. The association with *O. dioscoreae* is ubiquitous throughout the geographic range of *D. sansibarensis* and to our knowledge aposymbiotic *D. sansibarensis* are not found in nature [26], indicating a strong mutualistic interaction. Together, this indicates that the fitness benefit provided to the partners of the *D. sansibarensis*/*O. dioscoreae* may be contingent on environmental factors, such as biotic or abiotic stresses. Remarkably, the leaf glands of aposymbiotic plants left in non-sterile conditions may become colonized by bacteria other than *O. dioscoreae* (S3 Table). This indicates that the association may not be strictly controlled, or least that the mechanisms which control colonization of leaf glands are not sufficient to prevent opportunistic infections in the absence of *O. dioscoreae*. Whether opportunistic associations with bacteria other than *O. dioscoreae* are stable in a single host or across generations remains to be tested.

In conclusion, the ability to generate aposymbiotic *D. sansibarensis*, coupled with the ability to culture and genetically manipulate *O. dioscoreae*, provides an interesting opportunity to investigate vertically-transmitted symbioses in plants. To our knowledge, this is the only heritable plant symbiosis known where both host and symbiont can be grown separately and where the symbiont can be easily manipulated. Further exploiting this system could provide new insights into the evolution of heritable leaf symbiosis and vertically-transmitted symbioses in general.

## Supporting information

**S1 Fig. Method developed to make aposymbiotic plants and re-introduce a bacterium of interest. (A)** Node cuttings were taken from adult plants and incubated for 8 hours in 5% PPM for initial sterilization. **(B)** Node cuttings were incubated in a mixture of liquid MS, antibiotics and PPM for 3 weeks. **(C)** After 3–4 weeks, a bulbil (b) with its root system became apparent. Multiple leaves have formed from the node and are providing sugars to the plant. **(D)** The bulbil grows its own stem (s) that uses gravitropism to grow up and after the emergence of 2 leaves, the apical bud becomes visible. **(E)** After confirmation of being aposymbiotic by crushing and plating out the newly developed acumen(s), the plant was re-inoculated with a bacterium of interest by dropping 2 µl of the bacterial suspension on the apical bud.
(PDF)

**S2 Fig. Morphological parameters of aposymbiotic vs. symbiotic *D. sansibarensis* in gnotobiotic conditions.** Wild-type colonized *D. sansibarensis* were inoculated by a *O. dioscoreae* R-71412 cell suspension (Orrella) or a sterile 0.4% NaCl solution (MOCK) and grown for 4 weeks in gnotobiotic conditions. Leaf surface area (A) and length of the forerunner tip containing the bacterial glands (B) were measured for 3 leaves per plant, starting with the leaf closest to the shoot tip (leaf 1, not shown). C. Total stem length measured from the crown to the shoot tip. Data from 2 independent experiments are shown separately. Data from mock-inoculated plants are shown in orange, and in blue for *O. dioscoreae*-inoculated plants. The distributions of values between the *O. dioscoreae*–or mock-inoculated plants are identical for each of the 3 parameters (Wilcoxon rank sum test $p > 0.05$).
(PDF)

**S3 Fig. Physiological parameters of aposymbiotic vs. symbiotic *D. sansibarensis* in gnotobiotic conditions.** Wild-type colonized *D. sansibarensis* were inoculated by a *O. dioscoreae* R-71412 cell suspension (Orrella) or a sterile 0.4% NaCl solution (MOCK). Physiological parameters were measured using a hand-held optical meter after 4 weeks of growth in gnotobiotic conditions. Parameters measured include **A**. Chlorophyl content (Chl); **B**. Anthocyanins index, measured as a function of green light absorbed by the sample; **C**. Flavonoids index (Flav), measured as a function of UV light absorbed by the sample and **D**. Nitrogen Balance Index (NBI) is measured as the ratio of Chl and Flav and is an indicator of C/N allocation changes due to N-deficiency. Data from 2 independent experiments are shown separately. Data from mock-inoculated plants are shown in orange, and in blue for *O. dioscoreae*-inoculated plants. The distributions of values between the *O. dioscoreae*–or mock-inoculated plants are identical for each of the 4 parameters (Wilcoxon rank sum test $p > 0.05$).
(PDF)

**S4 Fig. Physiological parameters of aposymbiotic vs. symbiotic *D. sansibarensis* in gnotobiotic conditions.** Wild-type colonized *D. sansibarensis* were inoculated by a *O. dioscoreae* R-71412 cell suspension (Orrella) or a sterile 0.4% NaCl solution (MOCK). Physiological parameters were measured using a hand-held optical meter after 4 weeks of growth in gnotobiotic conditions. Parameters measured include **A**. Chlorophyl content (Chl); **B**. Anthocyanins index, measured as a function of green light absorbed by the sample; **C**. Flavonoids index (Flav), measured as a function of UV light absorbed by the sample and **D**. Nitrogen Balance Index (NBI) is measured as the ratio of Chl and Flav and is an indicator of C/N allocation changes due to N-deficiency. Data from 2 independent experiments are shown separately. Data from mock-inoculated plants are shown in orange, and in blue for *O. dioscoreae*-inoculated plants. The distributions of values between the *O. dioscoreae*–or mock-inoculated plants

are identical for each of the 4 parameters (Wilcoxon rank sum test $p > 0.05$).
(PDF)

**S1 Table. Bacterial species used in this study.**
(PDF)

**S2 Table. Minimum inhibitory concentrations of biocidal products on different *O. dioscoreae* strains.**
(PDF)

**S3 Table. Symbiotic status of plants used in phenotyping experiment.** APO = aposymbiotic status, SYM = symbiotic status, check-ups quantified the amount *of O. dioscoreae* found in new leaf acumens. Not = Majority isolates not identified as *O. dioscoreae*. Last column gives the eventual identity given to the sample for further analysis: APO = aposymbiotic plant, unknown = colonized by bacteria other than *O. dioscoreae*, *Orrella dioscoreae* = colonized by *Orrella dioscoreae*.
(PDF)

## Acknowledgments

We are grateful to Bart Panis (KU Leuven) and Danny Geelen (UGent) for very helpful discussions in the early stages of this study. We also thank Fabrice Devoilles and Marine Pinsard of the LIPME Plant production facilities, as well as Christian Pince, Florian Delhommeau and Dominique Senac for their help with sterilization protocols and media preparation. This study is set within the framework of the "Laboratoires d'Excellences (LABEX)" TULIP (ANR-10-LABX-41) and of the "École Universitaire de Recherche (EUR)" TULIP-GS (ANR-18-EURE-0019).

## Author Contributions

**Conceptualization:** Tessa Acar, Aurelien Carlier.

**Data curation:** Marie-Françoise Jardinaud.

**Formal analysis:** Marie-Françoise Jardinaud, Gabriella Houdinet.

**Funding acquisition:** Aurelien Carlier.

**Investigation:** Tessa Acar, Sandra Moreau, Gabriella Houdinet, Olivier Leroux, Olivier Coen.

**Methodology:** Tessa Acar, Sandra Moreau, Frédéric De Meyer, Bart Hoste, Aurélie Le Ru.

**Project administration:** Aurelien Carlier.

**Software:** Felicia Maviane-Macia.

**Supervision:** Nemo Peeters, Aurelien Carlier.

**Writing – original draft:** Tessa Acar, Aurelien Carlier.

**Writing – review & editing:** Aurelien Carlier.

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
