## [Decision Letter · Decision Letter 0]

19 Dec 2023

PONE-D-23-32436The association between *Dioscorea sansibarensis* and *Orrella dioscoreae* as a model for hereditary leaf symbiosisPLOS ONE

Dear Dr. Carlier,

Thank you for submitting your manuscript to PLOS ONE. After careful consideration, we feel that it has merit but does not fully meet PLOS ONE’s publication criteria as it currently stands. Therefore, we invite you to submit a revised version of the manuscript that addresses the points raised during the review process.

I am really sorry for this long wait. Your work was evaluated by two experts who gave very positive feedback, but also highlighted some flaws, so I would appreciate if you reviewed your manuscript accordingly

We look forward to receiving your revised manuscript.

Kind regards,

Clara F. Rodrigues

Academic Editor

PLOS ONE

“We are grateful to the TRI-FRAIB imaging platform facilities, FR AIB 3450 CNRS-UTIII member of the national infrastructure France-BioImaging supported by the French National Research Agency (ANR[1] 10-INBS-04). This work was supported by the UGent Special Research Fund under grant 21BOFSTA2017002001 to AC. AC also acknowledges support from the French National Research Agency under grant agreement ANR-19-TERC-0004-01 and from the French Laboratory of Excellence project "TULIP" (ANR-10-LABX-41; ANR-11-IDEX-0002-02). The funders had no role in study design, data collection and analysis, decision to publish, or preparation of the manuscript.”

We note that you have provided additional information within the Acknowledgements Section that is currently declared in your Funding Statement. Please note that funding information should not appear in the Acknowledgments section or other areas of your manuscript. We will only publish funding information present in the Funding Statement section of the online submission form.

“This work was supported by the UGent Special Research Fund under grant BOFSTA2017002001 to AC. AC also acknowledges support from the French National Research Agency under grant agreement ANR-19-TERC-0004-01 and from the French Laboratory of Excellence project  "TULIP" (ANR-10-LABX-41; ANR-11-IDEX-0002-02). The funders had no role in study design, data collection and analysis, decision to publish, or preparation of the manuscript.”

4. Please update your submission to use the PLOS LaTeX template. The template and more information on our requirements for LaTeX submissions can be found at http://journals.plos.org/plosone/s/latex.

6. We note that Figures 1 and Figure S1 in your submission contain copyrighted images. All PLOS content is published under the Creative Commons Attribution License (CC BY 4.0), which means that the manuscript, images, and Supporting Information files will be freely available online, and any third party is permitted to access, download, copy, distribute, and use these materials in any way, even commercially, with proper attribution. For more information, see our copyright guidelines: http://journals.plos.org/plosone/s/licenses-and-copyright.

1. You may seek permission from the original copyright holder of Figures 1 and Figure S1 to publish the content specifically under the CC BY 4.0 license.

Reviewers' comments:

Reviewer's Responses to Questions

**Comments to the Author**

1. Is the manuscript technically sound, and do the data support the conclusions?

Reviewer #1: Yes

Reviewer #2: Yes

2. Has the statistical analysis been performed appropriately and rigorously? 

Reviewer #1: Yes

Reviewer #2: I Don't Know

3. Have the authors made all data underlying the findings in their manuscript fully available?

Reviewer #1: Yes

Reviewer #2: No

4. Is the manuscript presented in an intelligible fashion and written in standard English?

Reviewer #1: Yes

Reviewer #2: Yes

5. Review Comments to the Author

Reviewer #1: Date: 10/25/23

Journal: Plos One

Manu #: PONE-D-23-32436

Title: The association between Dioscorea sansibarensis and Orrella dioscoreae as a model for hereditary leaf symbiosis

Authors: Tessa Acar; Sandra Moreau; Marie-Francoise Jardinaud; Gabriella Houdinet; Felicia Maviane-Macia; Frederic De Meyer; Bart Hoste; Olivier Leroux; Olivier Coen; Aurelie Le Ru; Nemo Peeters; Aurelien Carlier

General Comments: This study deals with understanding the leaf symbiotic relationship between the host, Dioscorea sansibsarensis, and its bacterium, Orrella dioscoreae. This symbiotic relationship is unique among other symbiotic leaf taxa in other plant families as the study shows the bacterium can be cultured outside the plant, the plant can grow normally without the bacterium, and the same bacterium or genetically modified bacterium can be reintroduced into the uninfected plants. This experimental regime satisfies Koch’s postulates (which should be mentioned in the text). This experimental system may allow for a better understanding of other similar or obligate leaf symbioses as to their functional value(s).

Specific Comments: there are comments made throughout the attached, edited text. The monocot Dioscorea sansibsarensis symbiotic relationship seems significantly different from the obligate dicot leaf symbioses in that the bacteria are always on the leaf surface covered by the curled leaf lamina tip and associated secretory trichomes, and never encapsulated internally like with the nodules of Psychotria, Pavetta, and Ardisia sps. The enclosed shoot apices of the latter sps. do serve as an ‘external’ reservoir with stipular secretory glands/trichomes that maintain a bacterial population until it penetrates into the leaf lamina to form encapsulated nodules. However, none of these relationships show the bacteria inside cells but remain intercellular within the isolated nodules, a completely different environment than in this study. Whether further research on the system in the present study will shed light on the former obligate taxa remains an open question. With this said, the present study is worthy of publication because future research may uncover information that is relevant to it and the obligate taxa mentioned.

Title: appropriate;

Abstract: Identifies the purpose of the study and its rationale;

Key words: no key words listed in body but shown previously;

Introduction: Provides an excellent background for why the study is being conducted. It covers the key references and points out its significance related to previous studies;

Material and methods: See various comments. the M&m are presented in great detail by individual sections, and can be readily reproduced by other researchers, if necessary;

Results: See comments throughout Results;

Discussion: covers the Results adequately. It would be useful to include adding how the present study compares to Koch’s postulates as all other leaf nodule ‘obligate symbioses fail to satisfy them. Furthermore, the functional value(s), relayed to them being obligate, is still in question, as is the association in the present study.

References: match those listed in text and are appropriate to this study, and useful to any researcher interested in this area of inquiry;

Table 1: clear results;

Table 2: clear results;

Fig. 1: see various comments made in the legend. Satisfying the comments should greatly improve the quality of the images;

Fig.: 2: see comment in legend;

Fig. 3: Clear results;

Fig. S1: detailed, clear, and ok. Images of good quality;

Fig. S2: detailed, clear, and ok

Fig. S3: detailed, clear, and ok

Table S1: detailed, clear, and ok

Table S2: detailed, clear, and ok

Table S3: detailed, clear, and ok

Reviewer #2: Review Accar 2023

Really nice work, you can see that even though it’s the most ‘easy’ interaction in

---

## [Author Response · Author response to Decision Letter 0]

25 Jan 2024

A point-by-point response to reviewer's comments was uploaded as a separate file.

---

## [Decision Letter · Decision Letter 1]

3 Apr 2024

The association between *Dioscorea sansibarensis* and *Orrella dioscoreae* as a model for hereditary leaf symbiosis

PONE-D-23-32436R1

Dear Dr. Carlier,

We’re pleased to inform you that your manuscript has been judged scientifically suitable for publication and will be formally accepted for publication once it meets all outstanding technical requirements.

Kind regards,

Clara F. Rodrigues

Academic Editor

PLOS ONE

Additional Editor Comments (optional):

Thank you for addressing all the comments and I am sorry for the time it took all the process 

Reviewers' comments:

Reviewer's Responses to Questions

**Comments to the Author**

1. If the authors have adequately addressed your comments raised in a previous round of review and you feel that this manuscript is now acceptable for publication, you may indicate that here to bypass the “Comments to the Author” section, enter your conflict of interest statement in the “Confidential to Editor” section, and submit your "Accept" recommendation.

Reviewer #2: All comments have been addressed

2. Is the manuscript technically sound, and do the data support the conclusions?

Reviewer #2: Yes

3. Has the statistical analysis been performed appropriately and rigorously? 

Reviewer #2: Yes

4. Have the authors made all data underlying the findings in their manuscript fully available?

Reviewer #2: Yes

5. Is the manuscript presented in an intelligible fashion and written in standard English?

Reviewer #2: Yes

6. Review Comments to the Author

Reviewer #2: Thank you for addressing all comments so rigorously!

The changes made it easier for the me as reader to follow the article.

I hope this article will help to further train future generation AI bots to formulate the function of leaf nodulating symbiosis better.

No further remarks on the text.

Good job!

7. PLOS authors have the option to publish the peer review history of their article (what does this mean?). If published, this will include your full peer review and any attached files.

Reviewer #2: No

---

## [Editor Report · Acceptance letter]

8 Apr 2024

PONE-D-23-32436R1 

PLOS ONE

Dear Dr. Carlier, 

I'm pleased to inform you that your manuscript has been deemed suitable for publication in PLOS ONE. Congratulations! Your manuscript is now being handed over to our production team.

Kind regards, 

on behalf of

Dr. Clara F. Rodrigues 

Academic Editor

PLOS ONE